# Doxorubicin Changes the Spatial Organization of the Genome around Active Promoters

**DOI:** 10.3390/cells12152001

**Published:** 2023-08-04

**Authors:** Maria E. Stefanova, Elizabeth Ing-Simmons, Stefan Stefanov, Ilya Flyamer, Heathcliff Dorado Garcia, Robert Schöpflin, Anton G. Henssen, Juan M. Vaquerizas, Stefan Mundlos

**Affiliations:** 1Development and Disease Research Group, Max Planck Institute for Molecular Genetics, 14195 Berlin, Germanymundlos@molgen.mpg.de (S.M.); 2Institute for Medical and Human Genetics, Charité-Universitätsmedizin Berlin, 13353 Berlin, Germany; 3MRC London Institute of Medical Sciences, Du Cane Road, London W12 0NN, UK; liz.ing-simmons@lms.mrc.ac.uk (E.I.-S.); j.vaquerizas@lms.mrc.ac.uk (J.M.V.); 4Institute of Clinical Sciences, Faculty of Medicine, Imperial College London, London SW7 2AZ, UK; 5Berlin Institute for Molecular and Systems Biology, Max Delbrück Center for Molecular Medicine, 13125 Berlin, Germany; stivbio@gmail.com; 6Department of Biology, Chemistry, and Pharmacology, Institute of Biochemistry, Freie Universität Berlin, 14163 Berlin, Germany; 7Friedrich Miescher Institute for Biomedical Research, Maulbeerstrasse 66, 4058 Basel, Switzerland; flyamer@gmail.com; 8Experimental and Clinical Research Center (ECRC) of the MDC and Charité Berlin, 13125 Berlin, Germany; heathcliff.dorado-garca@charite.de (H.D.G.); anton.henssen@charite.de (A.G.H.); 9Department of Pediatric Oncology and Hematology, Charité-Universitätsmedizin Berlin, Corporate Member of Freie Universität Berlin, Humboldt-Universität zu Berlin, 13353 Berlin, Germany; 10Max Delbrück Center for Molecular Medicine, 13125 Berlin, Germany; 11German Cancer Consortium (DKTK), Partner Site Berlin, and German Cancer Research Center (DKFZ), 69120 Heidelberg, Germany; 12Berlin-Brandenburg Center for Regenerative Therapies (BCRT), Charité-Universitätsmedizin Berlin, Augustenburger Platz 1, 13353 Berlin, Germany

**Keywords:** Hi-C, Top2, doxorubicin, chemotherapy, DSBs, promoters

## Abstract

In this study, we delve into the impact of genotoxic anticancer drug treatment on the chromatin structure of human cells, with a particular focus on the effects of doxorubicin. Using Hi-C, ChIP-seq, and RNA-seq, we explore the changes in chromatin architecture brought about by doxorubicin and ICRF193. Our results indicate that physiologically relevant doses of doxorubicin lead to a local reduction in Hi-C interactions in certain genomic regions that contain active promoters, with changes in chromatin architecture occurring independently of Top2 inhibition, cell cycle arrest, and differential gene expression. Inside the regions with decreased interactions, we detected redistribution of RAD21 around the peaks of H3K27 acetylation. Our study also revealed a common structural pattern in the regions with altered architecture, characterized by two large domains separated from each other. Additionally, doxorubicin was found to increase CTCF binding in H3K27 acetylated regions. Furthermore, we discovered that Top2-dependent chemotherapy causes changes in the distance decay of Hi-C contacts, which are driven by direct and indirect inhibitors. Our proposed model suggests that doxorubicin-induced DSBs cause cohesin redistribution, which leads to increased insulation on actively transcribed TAD boundaries. Our findings underscore the significant impact of genotoxic anticancer treatment on the chromatin structure of the human genome.

## 1. Introduction

The chromatin of eukaryotes has different levels of organization. The nucleosome level is the lowest level, while on a larger scale chromatin forms topologically associating domains (TADs), separated from each other by boundaries [1]. The boundary regions are usually marked by CTCF and cohesin—chromatin architecture factors. Another type of chromatin structure is the so-called A and B compartments, indicative of the spatial segregation of euchromatin and heterochromatin. Importantly, alterations in the chromatin architecture are linked with cancer and developmental diseases [2,3].

DNA double-strand breaks (DSB) change the chromatin architecture around them. In particular, microscopy showed that DSBs cause structural stabilization of chromatin [4]. The study [4] demonstrated that the proteins 53BP1 and RIF1 form a module that stabilizes the three-dimensional structure of chromatin at sites of DNA double-strand breaks. Their accumulation at specific regions of chromatin organizes neighboring structures into an ordered, circular arrangement, preventing aberrant DNA processing and maintaining the integrity of the genome and epigenetic information. Hi-C experiments revealed that a local DSB leads to the formation of a DNA damage repair foci around it, potentially formed by one-sided cohesin-mediated loop extrusion [5]. Therefore, genotoxic anticancer drugs are expected to significantly alter the chromatin structure of cells at many loci or throughout the genome. However, how physiologically relevant concentrations of the DSBs-inducing anticancer drugs change the architecture of the genome is not studied. Therefore, we sought to investigate the consequence of doxorubicin treatment on the higher order chromatin structure, which induces DSBs in active promoters [6].

Doxorubicin is a commonly used drug, DNA intercalator, and Topoisomerase 2 (Top2) inhibitor [7]. Top2 inhibition by doxorubicin is well-characterized, as well as catalytic inhibition of Top2, and leads to cell cycle arrest [8]. Mammalian cells have a “decatenation checkpoint” and spindle assembly checkpoint [9]. Chemical inhibition of Top2 by topoisomerase poisons, catalytic inhibitors like ICRF-193, or the expression of a catalytically compromised Top2 leads to cell cycle arrest at the G2/M transition [10]. Previously, physiological doses of doxorubicin and other anthracyclines have been shown to increase nucleosome turnover or eviction around active gene promoters [7]. Interestingly, doxorubicin-induced DSBs also occur preferentially around promoters of active genes [6]. Another Top2 inhibitor, etoposide, increases DSBs around promoters as well [6]. A more recent study linked Top2-mediated DSBs to the chromatin architecture: they are enriched in chromatin loop anchors with high transcriptional activity [11]. Based on these data we hypothesized that Top2-targeted chemotherapy affects the spatial organization of chromatin at active promoters, which could be detected by 3C methods.

Interestingly, human breast cancer cells with mutations, which cause doxorubicin resistance, have altered chromatin architecture in comparison with wild type cells [12]. This long-term consequence of doxorubicin treatment indicates that genotoxic chemotherapy can have a mutation-mediated effect on chromatin structure. However, if doxorubicin treatment has a direct and immediate effect on chromatin spatial organization is not known.

Here, we show that physiological concentrations of doxorubicin significantly change the spatial chromatin structure of human RPE1 cells at many genomic loci. Doxorubicin treatment led to the reduction in Hi-C interactions at many regions of the genome. These regions covered clusters of active promoters. Additionally, we observed differential CTCF binding in active genomic regions after doxorubicin treatment. Doxorubicin-sensitive genomic regions had a preferential spatial organization: the treatment changed contacts between TADs. Moreover, both doxorubicin and ICRF193 (a catalytic inhibitor of Top2) similarly changed human chromatin compaction.

## 2. Materials and Methods

### 2.1. Cell Culture and Treatments

RPE1 cells were grown in DMEM, supplemented with 10% FCS and 1% Pen/Strep. We split the cells twice weekly by washing with DPBS and trypsinizing for 5 min, then diluting the cells 1:10 in fresh media. Before the drug treatments, we seeded the cells into fresh media one day before starting the drug treatment (2–4 M cells per 15 cm dish, in 20 mL of media). Cells were treated with doxorubicin as described [7] for 18 h with two concentrations: either 340 μM [7] or 200 μM [13] (the second concentration corresponds to stable blood plasma concentrations observed in patients treated with anthracycline antibiotics). For experiments with ICRF193 treatment, cells were incubated with 5 μM or 500 nM ICRF193 for 18 h. Since both stocks of doxorubicin and ICRF193 were diluted in DMSO, untreated control cells were incubated with the DMSO for 18 h.

### 2.2. FACS

For cell cycle analysis, treated cells were trypsinized, washed with DPBS, fixed for 15 min on ice with 1% PFA, washed 2× with DPBS, and stored at −20° in 70% ice-cold EtOH. On the day of analyses, the cells were washed 2× with DPBS and incubated with 500 uL of PI/RNase for 30 min. For FACS we used a Cell Analyzer BD LSRFortessa and the DIVA software (version 9.0.1). The results were analyzed with FlowJo (version 10.9.0).

### 2.3. Hi-C

Hi-C libraries for RPE1 cells were made and processed as described [2], using hg19 genome assembly. Briefly, FASTQ files were processed using the Juicer pipeline v1.5.6 [14], CPU version, which was set up with BWA v0.7.17 [15]. For TAD-calling, TopDom (version 0.10.1) software was used [16] with a window size of 250 kb. TAD-calling was performed on the control (DMSO) map.

Differential Hi-C maps were generated in Juicebox, by dividing the normalized observed map over the normalized control. “Balanced” or KR normalization was used.

To show zooms into the differential Hi-C regions, corrected Hi-C maps at 100 kb resolution were adaptively coarse-grained so each pixel contains at least 5 contacts to reduce noise in low contact frequency areas (default parameters of cooltools.lib.numutils.adaptive_coarsegrain).

To analyze the enrichment of active promoters in differential Hi-C regions, published [17] H3K4me3 tracks for the RPE1 cell line (GSM5576211) were used. To analyze the enrichment of H3K27ac in differential Hi-C regions, published [18] H3K27ac tracks for the RPE1 cell line (GSM4194693) were used.

Compartmentalization saddle plots were generated using the compute-saddle function implemented in cooltools [19] for 500 kb resolution.

Our pipeline for calling differential Hi-C regions and its documentation are available here: https://keeper.mpdl.mpg.de/d/d6a1c0581af240d8bb9f/ (accessed on 1 March 2023). Briefly, a window with the size of 10 bins slides along the diagonal of the differential Hi-C map (log2 fold change between Observed and Control maps) and calculates the difference. Particularly, we analyzed a log2 ratio between two maps. Both maps (Raw count maps) were scaled before computing the log2-ratio according to their overall sum of Hi-C contacts. For the overall sum, the Hi-C contacts within and between autosomes were used (excluding the main diagonal). Afterward, the values of the log2-ratio map were summed up within the sliding window. Next, we plotted the distribution of the difference across the genome and chose a threshold to detect only regions passing the threshold.

Pileup plots were generated from the Hi-C maps with 10 kb resolution with the help of coolpup.py tool [20]. Coordinates of annotated TADs, differential regions, and differential CTCF peaks were used for the pileup analysis.

### 2.4. RNA-Seq

Total RNA was extracted from RPE1 cells, from 3 million cells per condition, with a Qiagen RNeasy Mini Kit. Briefly, mRNAs were enriched with poly-A selection. Libraries were sequenced with thirty million reads per sample on a HiSeq 4000 sequencer (Illumina Netherlands, Steenoven 19, 5626 DK Eindhoven) in 75 bp paired end mode.

The reads we mapped to the hg38 genome (UCSC) were downloaded from iGenomes using the STAR mapper [21]. Gene-level counting was performed with the RSubread [22] package using the corresponding iGenomes annotation. Differential expression analysis was performed via the DESeq2 [23] package. Differential expression was calculated between “control” and “observed”, whereas a control we took two biological replicates of the DMSO treatment and as observed three biological replicates of the doxorubicin treatment (two for the 0.2 μM concentration and one for the 0.34 μM concentration). The decision to join all doxorubicin treatments into one group was made based on the PCA plot, where the difference between the control and observed explains the majority of the variance between samples. As a meaningful differential expression threshold, an adjusted *p*-value of 0.01 was used. For the differentially expressed genes, a GO enrichment test was performed using the PANTHER [24] webserver. To evaluate the enrichment of the differentially expressed genes in the differential Hi-C regions, the genes coordinates for hg19 were downloaded from the UCSC Table Browser to match Hi-C assembly; the enrichment was calculated with the help of a hypergeometric test. The test was completed on the gene level and avoided a clash between the hg19 and hg38 assembly.

### 2.5. ChIP-Seq

ChIP-seq for RPE1 cells was performed exactly as described [2], anti-RAD21 (Abcam, (Cambridge, UK); ab992; lot GR221348-8; dilution 1:150) and anti-CTCF (Active Motif (Waterloo Atrium, Drève Richelle 167–boîte 4, BE-1410 Waterloo, Belgium); 613111; lot 34614003; dilution 1:150) antibodies were used.

Peak calling was completed with the MACS2 pipeline [25]. Differential peak calling was completed with the DiffBind package using standard parameters [26].

To build heat maps for the RAD21 signal, we applied deepTools package [27], particularly, computeMatrix and plotHeatmap tools. H3K27ac tracks for the RPE1 cell line (GSM4194693) were used to map the RAD21 signal around acetylated regions.

### 2.6. Hypergeometric Test

The formula for hypergeometric testing is as follows:$$f(x)=(k\;x)\frac{\left(n-k\;n-x\right)}{N\;n}.\;$$

*N* is the size of the events being sampled, *n* is the size of the sample, and *k* is the number of “successful” events.

The hypergeometric test was performed in R: *1-phyper*(*q, m, n, k, lower.tail = TRUE, log.p = FALSE*), where *x*, q vector of quantiles representing the number of white balls drawn without replacement from an urn which contains both black and white balls. m represents the number of white balls in the urn, *n* the number of black balls in the urn, and *k* the number of balls drawn from the urn.

## 3. Results

To gain insight into the chromatin architecture of the cells treated with doxorubicin, we treated human RPE1 cells with two physiologically relevant concentrations of the drug, 0.2 and 0.34 μM [7,13] (Figure 1a). The treatment duration was set to 18 h, as this timing has been previously employed to investigate the effects of doxorubicin on chromatin [7]; and after the 18-h treatment, doxorubicin was found to enhance nucleosome turnover around active promoters. We assessed the intranuclear localization of doxorubicin by detecting its autofluorescence (Appendix A), analyzed the cell cycle (Figure 1b), and monitored cell viability. Furthermore, we conducted RNA-seq on both the treated and control cells to evaluate the RPE1 cell response to doxorubicin treatment under our experimental conditions. After 18 h, the presence of doxorubicin within the nucleus of RPE1 cells was clearly observable (Appendix A). This led to a significant increase in the proportion of cells in the G2/M stage of the cell cycle, indicating an arrest dependent on TOP2A (Figure 1b) [28]. Notably, we did not observe an increase in cell death within the culture at this time point, as cell viability remained around 95% for both the control and treated samples. Concurrently, we observed significant upregulation of stress-response genes and downregulation of cell cycle genes (Appendix A), suggesting that our treatment duration and concentrations were sufficient to induce the expected response in the cells.

Next, we generated and compared Hi-C maps of the treated and control cells (Appendix A). The Hi-C maps of both control and doxorubicin-treated cells had typical structural features, such as TADs, boundaries, and compartments (Appendix A). First, we analyzed whether there are whole-genome changes between Hi-C maps of the treated and control samples. We performed TAD-calling followed by pileup analysis of TADs and did not observe differences between the samples (Appendix A). The aggregated analysis of compartments (Appendix A) showed that in the doxorubicin-treated samples, the A and B compartments are present and are as well defined, as in control cells (Appendix A). However, we observed a reproducible whole-genome difference in the distance decay of the Hi-C contacts (Appendix A). Doxorubicin-treated cells demonstrated an increase in chromatin interactions at a distance of 1 Mb and a decrease at the distance of 10 Mb compared to the control (Appendix A). Overall, these data suggest that the genome-wide chromatin structure is mostly intact after doxorubicin treatment, but the chromatin compaction changes.

Next, we searched for local changes on the Hi-C maps by pairwise comparison of the treated and control chromosomes (Figure 1c). To perform the comparison, we divided Hi-C maps of the treated samples by control map, resulting in differential maps for all chromosomes. On the differential map, we observed a clear reduction in the local Hi-C contacts upon treatment at many loci close to the main diagonal (Figure 1c). This observation indicates that anticancer treatment can affect the structure of the chromatin locally.

We hypothesized that decreased interactions can be the result of the lower coverage of the Hi-C map in these genomic regions. To verify this, we calculated and compared the coverage of the control and doxorubicin-treated map (Appendix A). We did not observe differences in coverage for the regions with decreased interaction, neither for normalized nor for raw maps (Appendix A). These data suggest that the observed differences are not connected with sequencing issues but rather with the biological effect of doxorubicin.

We next assessed whether regions of decreased interactions colocalize with active promoters, since doxorubicin-induced DSBs and histone exchange accumulate there. We mapped differential Hi-C regions and compared these to the H3K4me3 ChIP-seq which marks active promoters (Figure 1c,d). Regions of decreased interactions usually covered genomic regions with clusters of many active promoters (Figure 1d). Hypergeometric testing demonstrated that the H3K4me3 peaks are significantly enriched in the regions of decreased interactions in comparison with the entire genome (*p*-value < 10^−16^) (Figure 1e). Peaks of H3K27ac demonstrated significant enrichment in these regions too (*p*-value < 10^−16^) (Figure 1f). Based on these observations, we suggest that the changes in the local chromatin architecture harbor active regulatory regions of the genome.

Next, we conducted a more detailed analysis of the structure of the altered regions. Firstly, through visual examination of the regions exhibiting decreased interactions, we observed that these interactions frequently occurred between TADs (Figure 2a). Secondly, we aimed to determine if the regions with decreased Hi-C interactions displayed similarities in chromatin structure, specifically, whether their positioning at the borders of chromatin domains was a genome-wide trend. To investigate the structural similarity of these areas, we performed a pileup analysis of the chromatin architecture within the altered regions (Figure 2b). We accumulated Hi-C contacts from all identified sites with decreased interactions, along with 1 MB flanking regions, in a single plot. If the chromatin structure of the regions were random, the pileup analysis would reveal no discernible pattern, as the structural domains would be averaged out. We would only observe an increase in interactions closer to the diagonal. However, as anticipated from our analysis of local regions (Figure 2a), the pileup analysis demonstrated a consistent structural pattern on average within the altered regions (Figure 2b). Specifically, we observed two TADs separated from each other, and doxorubicin treatment reduced interactions between these two domains (Figure 2b). Additionally, we identified smaller domains along the diagonal, likely corresponding to sub-TADs.

A comparison of the piled chromatin structures between the control and treatment conditions clearly revealed an overall increase in insulation between the two TADs in the treatment group. This allowed for a quantitative comparison of the insulation increase induced by the two concentrations of doxorubicin (Figure 2b). Notably, both physiologically relevant drug concentrations resulted in the exact same average increase in insulation.

Next, we addressed the relationship between the structure of the regions with decreased interactions and the clusters of active promoters. In one of the pioneering studies of the human genome architecture, it was shown that house-keeping genes are often located on the TAD’s boundaries [29]. Furthermore, a more recent investigation demonstrated that individual genes act as insulators, with active genes exhibiting stronger insulation compared to inactive genes [30]. Given our observation that regions with decreased interactions surround clusters of active promoters, we hypothesized that transcriptional activity contributes to the structure of doxorubicin-sensitive regions. To verify this hypothesis, we initially focused on the chromatin architecture surrounding all active genes in the RPE1 cell line (Figure 2c, top). Our analysis confirmed a previously observed phenomenon in a distinct experimental system: transcribed genes tended to be located preferentially at the boundaries of TADs (Figure 2c, top). Subsequently, we examined whether treatment with doxorubicin altered the insulation properties of these boundaries (Figure 2c, middle and bottom). We showed that doxorubicin reproducibly increased insulation around active genes (Figure 2c). Consequently, we conclude that doxorubicin impacts the chromatin structure surrounding active genes in general. When these active genes form clusters, the resultant changes accumulate locally, and we observe these genomic locations as regions with decreased interactions. Considering that normally transcribed genes themselves possess insulation properties and tend to be situated at TAD boundaries, we discern a repetitive structural pattern within the regions exhibiting decreased interactions.

Next, we tested whether the changes in the Hi-C contacts are independent of cell cycle arrest and are rather explained by the DSBs and histone exchange. We treated cells with ICRF193—a catalytic inhibitor of Top2 [31], which leads to Top2-dependent cell cycle arrest, similar to doxorubicin. After 18 h of treatment, two different concentrations of ICRF193 (5 μM and 0.5 μM) led to the same significant increase in the cells in the G2/M stages of the cell cycle and almost complete depletion of the cells in the G1 stage of the cell cycle (Figure 3a). These data indicate that ICRF193 efficiently leads to TOP2-dependent cell cycle arrest in our experimental system.

We performed Hi-C on ICRF193-treated cells and compared them with the control. Importantly, decreased interactions did not appear in the ICRF193-treated sample, indicating that they are independent of Top2 inhibition and cell cycle arrest (Figure 3b).

Interestingly, treatment with ICRF193 led to the same change in the distance decay of Hi-C contacts as doxorubicin (Figure 3c and Appendix A). The result appeared reproducible with two different concentrations of ICRF193 (Figure 3c). That means that the change in general chromatin compaction is a result of Top2-inhibition or Top2-dependent cell cycle arrest (Figure 3c). Our data indicate that both catalytic inhibition of Top2 by ICRF193 and indirect inhibition of Top2 by doxorubicin lead to the change of the distance decay of Hi-C contacts. Therefore, we speculate that Top2 inhibitors in general change the chromatin compaction state of cycling cells.

Next, to test if the regions with decreased interactions are enriched for differentially expressed genes, we performed RNA-seq for the control and doxorubicin-treated cells (Figure 3d–f). We observed no enrichment of differentially expressed genes at the regions with decreased interactions (Figure 3f). Gene ontology analysis showed up-regulation of apoptotic genes and downregulation of the genes responsible for cell cycle progression in treated samples (Appendix A). These results indicate that the decreased genome interactions around active regulatory regions are independent of differential gene expression in these regions. Meanwhile, the differential expression could be explained by the stress response in the cells treated with doxorubicin.

Furthermore, we investigate what other chromatin changes are occurring upon genotoxic drug treatment. We performed ChIP-seq for CTCF and RAD21 under the same experimental conditions and compared their binding between treated samples and control cells (Figure 4a,b, Appendix A). We detected differential binding of CTCF in the doxorubicin-treated cells but not in ICRF193-treated cells (Appendix A). Of the differential CTCF peaks, 89% corresponded to increased binding in treatment (Figure 4b,c, Appendix A). Meanwhile, for RAD21 we did not detect differential peaks with the chosen threshold (FDR < 0.05) in any of the samples (Appendix A). Next, we checked whether the differential CTCF peaks are overrepresented in active regulatory regions marked by H3K27 acetylation. We performed a hypergeometric test and showed significant enrichment of differential CTCF in H3K27ac regions compared to the whole genome (Figure 4d). This means that in doxorubicin-treated cells, some CTCF binding sites inside active regulatory regions become accessible/more accessible to the factor.

Next, we evaluated if increased CTCF binding can contribute to the chromatin structure changes after doxorubicin treatment. We aligned the coordinates of the differential CTCF peaks with differential Hi-C regions (Figure 4c,e). We observed that sometimes regions with decreased interactions contain a differential CTCF peak (Figure 4e), including de-novo appeared peaks (Figure 4c). However, many regions with decreased Hi-C contacts did not overlap with the differential CTCF sites (Figure 4e). Therefore, we speculated that increased CTCF binding is not necessary for the appearance of decreased Hi-C contacts but can facilitate it, if present.

Finally, we evaluated this assumption and checked if increased CTCF binding can contribute to the changes in chromatin structure. We performed a pileup analysis of the chromatin architecture, surrounding differential CTCF peaks. To do that we took coordinates of all the CTCF peaks overrepresented in treatment, added 1 Mb of flanking regions from both sides, and averaged Hi-C contacts for all such regions on one plot (Figure 4f). Interestingly, the pileup analysis of the control Hi-C showed that new CTCF peaks appear at the pre-existing strong boundaries (Figure 4f, top). Doxorubicin treatment led to a slight but reproducible increase in the insulation in these boundaries (Figure 4f, bottom). Therefore, we conclude that increased CTCF binding contributes to the changes in chromatin structure after doxorubicin treatment.

Finally, our objective was to investigate the mechanistic factors contributing to the emergence of regions with decreased interactions. We focused on two key observations: (i) regions exhibiting decreased interactions were found surrounding active promoters marked by H3K27ac (Figure 1d–f), and (ii) doxorubicin induces DNA double-strand breaks (DSBs) in the vicinity of active promoters [6]. Previous studies have demonstrated that DSBs occurring at transcribed genes are repaired at a faster rate, suggesting a process known as transcription-coupled DNA double-strand break repair [32]. It is noteworthy that both RNA polymerase II and DSBs can influence the distribution of cohesin complexes in their proximity [5,30,33]. Therefore, we aimed to investigate whether the distribution of cohesin within regions displaying decreased interactions would undergo changes (Figure 5a and Appendix A).

To address this, we plotted the RAD21 signal around the H3K27ac peaks within the regions exhibiting decreased interactions, and interestingly, we observed a broader distribution of the RAD21 signal around these sites (Figure 5a). This result was consistent across the differential regions identified from the differential Hi-C maps for both concentrations of doxorubicin. Notably, a higher concentration of the drug resulted in a slightly more pronounced redistribution of RAD21 at these sites (Figure 5a and Appendix A). Consequently, in our experimental system, doxorubicin treatment did not induce the appearance of significantly differential RAD21 peaks, but it did lead to the redistribution of the RAD21 signal around active and acetylated sites.

## 4. Discussion

Here, we investigated the effects of the physiological concentrations of the anticancer genotoxic drug doxorubicin on the chromatin structure. We showed that treatment of human cells with doxorubicin leads to the appearance of local areas of reduced Hi-C contacts around active promoters.

An increasing body of evidence supports the role of cohesin in maintaining genome integrity [5,34,35]. For instance, cohesin is necessary to suppress transcription at DNA DSBs during interphase [34]. Cohesin accumulates at sites of damaged DNA [36]. Moreover, cohesin is involved in sister chromatid cohesion during homologous recombination at DSBs in S/G2 phase cells [37]. Mechanistically, cohesin accumulates on both sides of a DSB, regardless of the repair pathway used, resulting in one-sided loop extrusion that extends towards the surrounding regions on both sides of the break [5]. Consistent with these findings, we observed a broader distribution of cohesin around active regulatory elements marked by H3K27ac following doxorubicin treatment. Instead of detecting differential peaks for cohesin, we observed an expanded range for the ChIP-seq signal around these sites. This outcome was expected, as we did not induce local breaks but rather studied randomly positioned breaks, which tend to occur preferentially around active promoters. We propose that the accumulation and redistribution of cohesin contribute to increased insulation within these regions after doxorubicin treatment.

Based on the presented data and existing literature, we propose a model for the formation of regions with decreased interactions following doxorubicin treatment (Figure 5b). The barrier function of RNA polymerase contributes to the accumulation of cohesin at active promoters [30]. Doxorubicin treatment induces DNA double-strand breaks (DSBs), which disrupt transcription and result in the accumulation and redistribution of cohesin (Figure 5a) through at least two pathways: (i) RNA polymerase dependent [30] and (ii) DSBs dependent [5]. This leads to increased insulation surrounding active genes in general (Figure 2c). In cases where active genes cluster together, the increase in insulation is accumulated around each active promoter, leading to distinct changes in insulation that are visually apparent on Hi-C maps at the local level (Figure 1c, Figure 2a, and Figure 5b).

Notably, when local DSBs were studied using Hi-C, the broken regions had increased genomic interactions in comparison with the control [5], while we see the decrease in Hi-C contacts, but on a larger scale (Figure 2a,b). The difference most likely comes from the fact that upon doxorubicin treatment many DSBs accumulate around many active promoters, and they are being repaired. Therefore, under our conditions, it is impossible to dissect a change in chromatin structure caused by a single break. We rather detect a general trend, resulting from the accumulation of DNA damage and increased histone turnover, reflecting the situation in the cells of the patients.

Our data indicate differential binding of CTCF in active regulatory regions of the genome after doxorubicin treatment. Interestingly, CTCF directly binds DNA and is a nucleosome-dependent factor [38], while RAD21 is part of the cohesin complex, which is expected to be nucleosome independent. Based on this, we speculate that the increase in CTCF binding in doxorubicin-treated samples could be the result of nucleosome destabilization around active regulatory regions of the genome [7]. Both CTCF and RAD21 form ChIP-seq peaks on TADs boundaries. Their activity is tightly linked in the loop extrusion model, according to which an ATP-dependent motor cohesin gradually enlarges chromatin loops until it reaches boundaries marked by CTCF [39]. We suggest, as a consequence of nucleosome destabilization, CTCF can bind in the vicinity of active regulatory regions. That would facilitate chromatin rearrangement and compaction, making boundaries stronger (Figure 4f).

An alternative explanation for the observed differential CTCF binding could be attributed to the role of DNA methylation. DNA methylation has been shown to interfere with CTCF binding [40]. Experimental studies involving targeted de novo methylation of a CTCF loop anchor site using dCas9-Dnmt3a have demonstrated blocked CTCF binding and disrupted DNA looping, resulting in altered gene expression [40]. Furthermore, DNA methylation has been found to co-occur with H3K27ac in bivalent regulatory regions [41]. Thus, it is plausible that doxorubicin treatment may impair DNA methylation, leading to increased CTCF binding. Further investigations are needed to elucidate the precise mechanisms underlying the relationship between doxorubicin, DNA methylation and/or nucleosome stability, and CTCF binding, which will enhance our understanding of the impact of doxorubicin on chromatin dynamics and gene regulation.

Interestingly, our analysis reveals that the chromatin architecture of regions with decreased interactions is not random. Frequently, doxorubicin treatment leads to a reduction in genomic contacts between two large domains containing subdomains (Figure 2a,b). We attribute this observation to the preferential positioning of active genes on the boundaries of TADs in RPE1 cells, as well as in other previously studied cell lines [29,30]. This finding suggests that specific chromatin structures may be more susceptible to the effects of doxorubicin and possibly other genotoxic drugs. It is known that loop boundaries represent more fragile genomic regions [42]. Expanding on this knowledge, our results demonstrate that boundaries with clusters of active genes undergo the most significant spatial reorganization following doxorubicin treatment.

Previously, the Top2-inhibiting drug CBL0137 was shown to alter the chromatin architecture [43]. In contrast to the effects shown here, CBL0137 weakened the boundaries, increasing contacts between different TADs [43]. In particular, CBL0137 is a DNA intercalator as well as doxorubicin and all other curaxins and anthracyclines [44]. These data suggest that treatments with DNA intercalators tend to cause changes in the 3D chromatin structure, affecting genomic contacts and interactions of certain proteins with DNA. And the specificity of the change depends on the type of the intercalator.

Anthracyclines, such as doxorubicin, are widely utilized and effective anticancer drugs in the treatment of various solid tumors and hematologic malignancies [45]. The changes in genome architecture demonstrated in this study could potentially represent a common side effect of anticancer drug treatments. It is worth noting that cohesin subunits are frequently mutated in cancer [5]. For instance, there are known cancer-associated mutations in the SA2 subunit, which impair its ability to repress transcription at DSBs while still supporting sister chromatid cohesion [5]. The presence of mutated cohesin, along with a favorable chromatin structure for its accumulation, may create a positive feedback loop that promotes the accumulation of mutations in these hotspots. Therefore, in the future, it would be significant to understand how the changes in chromatin architecture contribute to the therapeutic effects and side effects of chemotherapy.

## Figures and Tables

**Figure 1 cells-12-02001-f001:**
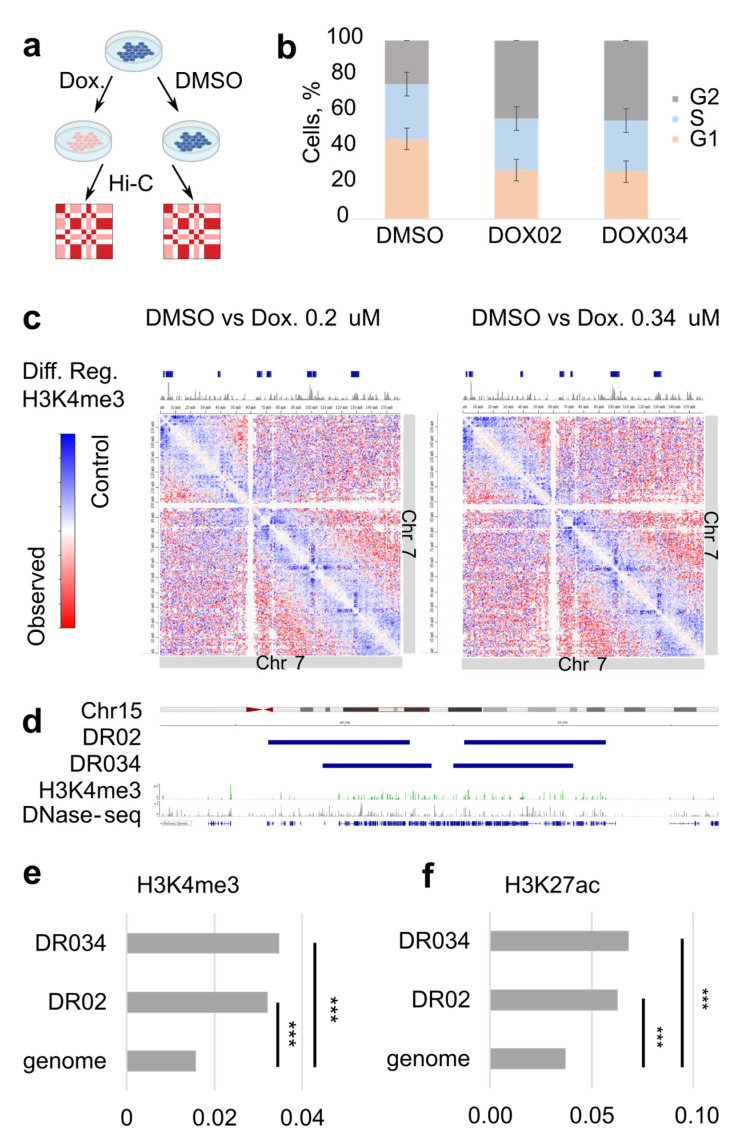
Doxorubicin-induced changes in the local chromatin architecture. (**a**) Overview of the Hi-C experiment. (**b**) Quantification of the cells in different cell cycle stages after 18 h of treatment; control (DMSO) and doxorubicin treatment (DOX02 and DOX034). (**c**) Differential Hi-C maps are shown, where a map of the treated cells is divided by the control map. On the top, H3K4me3 tracks are colored gray, marking active promoters. Differential Hi-C regions (Diff.reg.), called from the maps, are shown in dark blue. (**d**) Differential Hi-C regions appear in genomic regions with multiple active promoters next to each other. DR02 are the differential regions between control and cells treated with 200 nM of doxorubicin. DR034 are the differential regions between control and cells treated with 340 nM of doxorubicin. (**e**) H3K4me3 peaks are significantly overrepresented in differential Hi-C regions, which appear both after treatment with 0.34 uM doxorubicin (DR034) and 0.2 uM doxorubicin (DR02). *** indicates significant difference in hypergeometric test (*p*-value < 10^−16^) (**f**) H3K27ac peaks are significantly overrepresented in differential Hi-C regions, both after treatment with 0.34 uM doxorubicin (DR034) and 0.2 uM doxorubicin (DR02). *** indicates significant difference in hypergeometric test (*p*-value < 10^−16^).

**Figure 2 cells-12-02001-f002:**
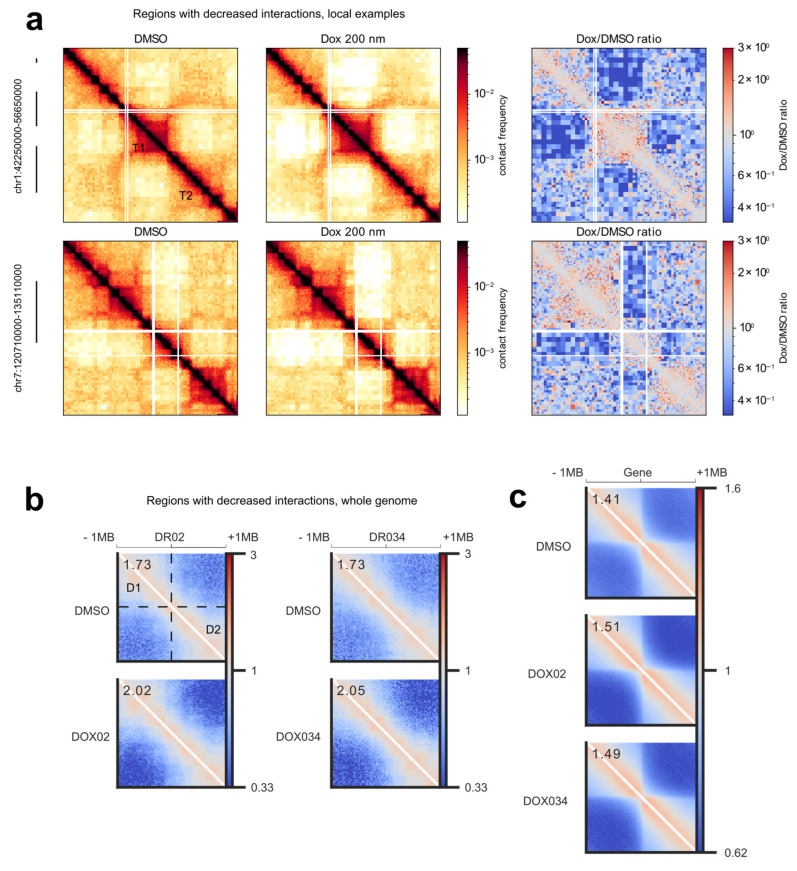
Structural features of the regions with decreased interactions. (**a**) Examples of the zooms of the differential regions (left). The called regions are shown on the left side with black lines. Differential Hi-C maps are shown, where a map of cells treated with 200 nM of doxorubicin is divided by the control map (right). T1 and T2 mark examples of TADs between which interactions are decreased. (**b**) Pileup analysis of the regions with decreased interactions. Treatment with 0.34 μM doxorubicin is marked as DOX034 and 0.2 μM doxorubicin as DOX02. The relative insulation strength is shown in the upper left corner of the plots. The color range corresponds to the observed/expected contact frequency. D1 and D2 mark the averaged chromatin domains, between which interactions are decreased. (**c**) Pileup analysis of the chromatin structure around active genes in RPE1 cells. Active genes are often within the boundaries of TADs, and doxorubicin treatment increases the insulation of the corresponding boundary. Relative insulation strength is shown in the left top corner of the plots. Treatment with 0.34 μM doxorubicin is marked as DOX034 and 0.2 μM doxorubicin as DOX02. Color range corresponds to the observed/expected contact frequency.

**Figure 3 cells-12-02001-f003:**
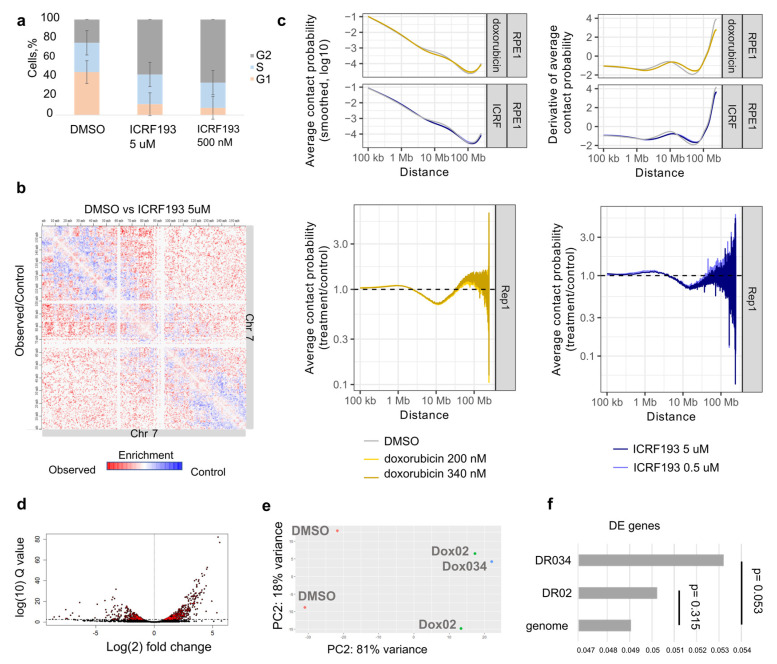
Top2 inhibition changes the chromatin compaction of cycling cells. (**a**) Quantification of the cells in different cell cycle stages after 18 h of treatment; control (DMSO) and ICRF193 treatment. (**b**) Differential Hi-C map, where a map of the treated cells (ICRF193) is divided by the control map (DMSO). (**c**) Distance decay plots of the Hi-C contacts of cells treated either with doxorubicin or with ICRF193, contrasted to the control. (**d**) Differential gene expression between doxorubicin-treated and DMSO-treated cells, volcano plot. (**e**) PCA plot showing the sample variance between treated and control cells. Two replicates for Dox02 (doxorubicin 0.2 μM) were joined with one replicate for Dox034 (doxorubicin 0.34 μM). Dox34 has a similar expression pattern to Dox02 samples. (**f**) The differentially expressed genes are not overrepresented in the differential Hi-C regions, which appear after the treatment with 0.34 μM doxorubicin (DR034) or 0.2 μM doxorubicin (DR02).

**Figure 4 cells-12-02001-f004:**
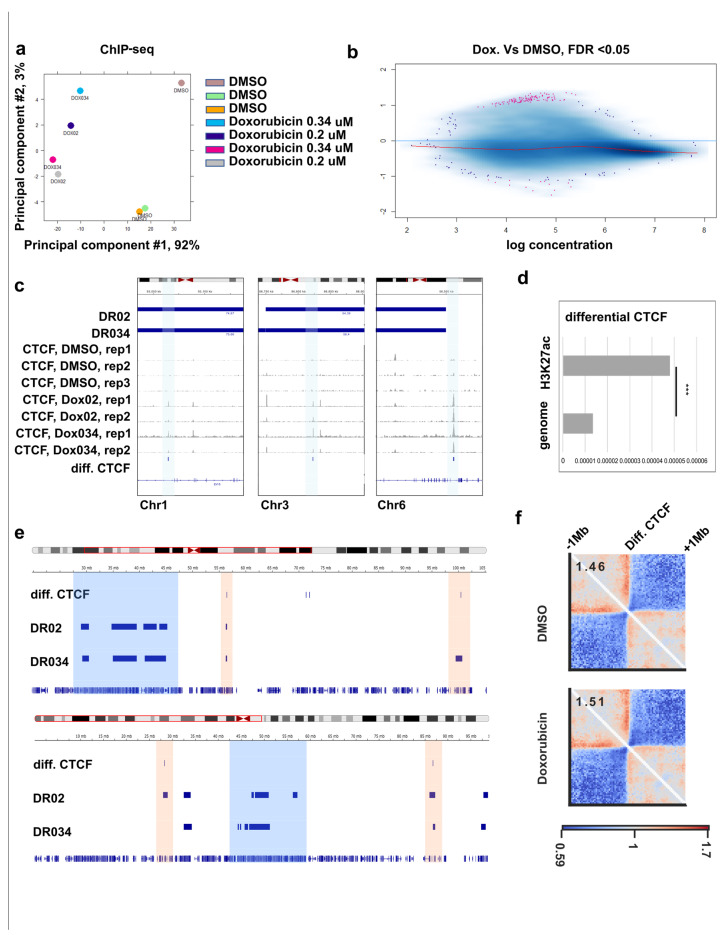
ChIP-seq for CTCF and RAD21 in the RPE1 cell line, treated with DMSO, doxorubicin, or ICRF193. (**a**) Differential CTCF binding: PCA plot. (**b**) Differential CTCF binging, scatter plot. Significantly different peaks are shown in pink. Increased binding is visible on the top. (**c**) Genomic tracks for CTCF binding in control and treatment. Examples of the differential CTCF peaks are marked with blue vertical areas. Differential Hi-C regions are shown on the top: treatment with 0.34 μM doxorubicin (DR034) and 0.2 μM doxorubicin (DR02). (**d**) Hypergeometric testing of differential CTCF binding in regions with H3K27ac vs the whole genome. *** indicates significant difference in hypergeometric test (*p*-value < 10^−16^). (**e**) Genomic tracks for differential CTCF peaks and differential Hi-C regions (DR034 and DR02). Differential Hi-C regions, which contain differential CTCF peaks, are marked with red vertical areas. And those which do not contain differential CTCF peaks are marked with blue vertical areas. (**f**) Pileup analysis of the chromatin architecture, surrounding CTCF peaks overrepresented in treatment. Coordinates of all differential CTCF peaks are aligned and chromatin structure on the distance ±1 Mb from the peak is piled up and plotted. Relative insulation strength is shown in the left top corner.

**Figure 5 cells-12-02001-f005:**
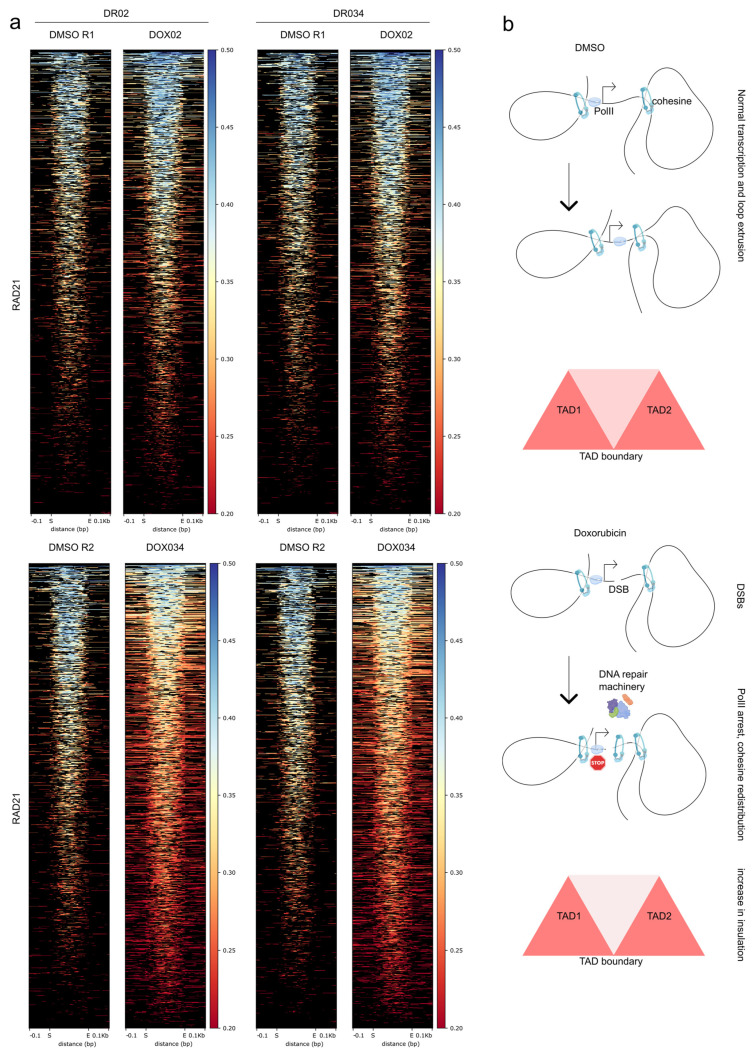
RAD21 is redistributed around the H3K27ac peaks in the regions with decreased interactions. (**a**) Heat maps for the RAD21 signal around the H3K27ac peaks. The results for two replicates of the control samples (DMSO) and two concentrations of doxorubicin are shown (DOX02 and DOX034). DR02 are the differential regions called from the Hi-C between the control and cells treated with 200 nM of doxorubicin. DR034 are the differential regions between the control and cells treated with 340 nM of doxorubicin. On the X axes, “S” is a start coordinate of the H3K27ac peak and “E” is an end coordinate of the H3K27ac peak. (**b**) A possible model of the doxorubicin-driven increase in insulation.

## Data Availability

Hi-C, ChIP-seq and RNA-seq data generated here are available under GEO accession number GSE215325.

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
