# Peer review of "Doxorubicin Changes the Spatial Organization of the Genome around Active Promoters"

_cells, 2023, doi:10.3390/cells12152001_

Round 1

Reviewer 1 Report (Previous Reviewer 1)

The authors have addressed the issues I raised regarding their first submission. 

Reviewer 2 Report (Previous Reviewer 2)

I think that the authors of the manuscript have performed great and extensive work revising the manuscript. I appreciate the more detailed explanations of the experimental rationale, the included cell cycle analysis and additional analyses of the results, the improved discussion with the model that explains doxorubicin effects on genome architecture. I would like to recommend this interesting work for publication in Cells.

This manuscript is a resubmission of an earlier submission. The following is a list of the peer review reports and author responses from that submission.

Round 1

Reviewer 1 Report

Stefanova et al. examined impact of doxorubicin on spatial genome organization in human cells and found evidence that doxorubicin reduces Hi-C interactions in some regions containing active promoters and increases CTCF binding in regions with H3-K27 acetylation. Their results presented another example of a DNA-interacting/intercalating anticancer drug impacting the 3D genome structure.  

Yang et al. has previously shown that doxorubicin enhances nucleosome turnover around promoter of active genes.  The authors speculated, but didn’t test, the relationship between nucleosome turnover and reduction in Hi-C interaction caused by doxorubicin. 

Kantiddze et al. recently found that another DNA-interacting/intercalating anticancer drug curaxin also weakens nucleosome stability like doxorubicin, but impacts spatial genome structure differently than doxorubicin.  Stefanova et al. didn’t provide any possible explanation for this discrepancy. Curaxin was shown (by Kantiddze et al.) to disrupt enhancer-promoter communications as a result of its effect on spatial genome organization, which is needed for expression of certain oncogenes, thereby providing a model for the anticancer activity of the drug. On the other hand, how doxorubicin induced changes in 3D genome structure is related to its anticancer function was not investigated or discussed by Kantiddze et al. here.  As such, it’s farfetched to state that this report would “provide insights that could aid in the development of more effective cancer treatments.”

Minor points

Line 168           Fig. 1a should be Fig. S1a

Line 357           There is no Fig. 2c

Line 382           change “in cell-cycle…” to “in a cell cycle…”

Reviewer 2 Report

In their article “Doxorubicin changes the spatial organization of the genome around active promoters” Stefanova et al. describe the effects of two genotoxic drugs, doxorubicin and ICRF193, on genome organization and chromatin compaction using Hi-C and complementing their data with Chip-seq and RNA-seq. The topic is certainly interesting and relevant. The limitations of the study are that it is purely descriptive and, in my view, superficial. Although I do not doubt the quality of the obtained data and the analysis of the results, I see several controversial and confusing points in both the experimental flow and the interpretation of the results. Therefore, I cannot recommend this work for publication in Cells.

Major points:

1.       RPE1 cells were used in the study and treatment with both drugs was performed for 18 hours. Why was this treatment time chosen and why only one treatment time? There is no description of the cells and how they are affected by the treatments. It would be more relevant to use several time points of treatment to compare short-term and long-term effects or at least justify the use of a single time point.

2.       Related to my previous comment, the authors check whether the changes in the Hi-C contacts are related to the Top2-dependent cell cycle arrest that the two drugs cause and conclude that some changes are dependent and some are not dependent. OK, do the treated cells undergo cell cycle arrest after 18 hours of treatment or not? Even if it is supposed to be obvious that they do, such data is not shown in the paper.

3.       Also, the authors conclude that inhibition of Top2 leads to the change of the distance decay of the Hi-C contacts. Where is the proof that Top2 was indeed inhibited by the drugs after 18 hours? I understand that this is the mechanism, by which these drugs actually work but it needs to be shown to which extent they indeed exert their effects in this particular experimental system.

4.       The whole paragraph about HAP1 cells looks artificially incorporated into this work. Given that this is a different cell system and considering my previous comment, I do not think that this part adds anything to support the authors’ conclusions. Additionally, it is also very confusing as the authors do not mention treating these cells with the studied drugs so Top-2A must be active in those cells. The conclusion that Top2A inhibition is causing some chromatin changes in RPE1 treated cells because Top2B is not causing them in HAP1 cells is not serious.

5.       Figure 2 is not clear. First, what are these “chromatin domains” and could you possibly mark them so it gets clear that interactions are decreased between them? Where are the two large domains in Figure 2b? Are the differential regions marked in green in Figure 2a? What exactly is the scale on Figure 2b? The legend could be more informative.

6.       The description of the results related to figure 2 is also confusing. The authors observe decreased interactions in the regions between chromatin domains (whatever this is) and then try to determine whether these have something in common and find out that the regions between domains have two domains…I simply fail to understand it all and to see any significance in this information, especially in view of the previous paragraph where it is already stated that decreased interactions colocalize with active promoters. What is the major point you want to make or the major conclusion you draw from this analysis? The conclusion in lines 236-239 repeats your previous conclusion that there are decreased interactions in treated cells. What is the relationship between the large chromatin domains and the clusters of active promoters?

7.       In the paragraph concerning RNA-seq the authors state that “regions with decreased interactions are not significantly enriched for differential gene expression” and then conclude that “changes in the chromatin architecture around active regulatory regions are independent of cell cycle arrest and differential gene expression”. First, why is the cell cycle arrest again mentioned here and second, what is the causal relationship the authors see – is it that changes in chromatin compaction should cause changes in gene expression or the other way round?

8.       There is no proper discussion of the results. The Discussion basically repeats the results of the study and does not offer any ideas, even speculative ones, how decreased interactions at active promoters and increased chromatin compaction could be related to the outcome of chemotherapy, except that they “could be a widespread side effect”. I understand that this work is focused on a particular aspect of doxorubicin’s effects on cells but even so some relationships need to be drawn. There are numerous sentences in the whole text that simply sound “empty”, e.g.:

“That could facilitate chromatin rearrangement and compaction, making boundaries stronger” – chromatin rearrangement is mentioned for the first time, what is it?

“This observation suggests that certain chromatin structures can be more easily affected by chemotherapy than others” – this does not sound very unexpected. Again, what are these chromatin structures?

Minor points:

1.       The first sentence in the introduction sounds weird since this paper is not focused on nucleosomes.

2.       Could you explain what is meant under “structural stabilization of chromatin” (line 50)?

3.       The sentence “Based on this data we hypothesized that Top2-related chemotherapy affects the spatial organization of chromatin at active promoters” suggests that the authors propose a completely novel idea, which is not true considering the papers they refer to in the preceding paragraph.

4.       In the last paragraph of the Introduction section these is another statement, which sounds unclear: “Unexpectedly, doxorubicin-sensitive genomic regions had a preferential spatial organization”. Why is it unexpected and what is this preferential organization?

5.       Figure 1a certainly does not represent an overview of the experiment as stated in the legend to this figure and I honestly do not see the relevance of it being shown at all.

6.       There is no figure 2c, which is referred to in line 375 of the Discussion section.